# Liquid Science and Digital Transformation: How Knowledge between Researchers Flows in Their Scientific Networks

**Simone Belli** [1,*] and **Ernesto Ponsot** [2]

1 Faculty of Political Science and Sociology, Complutense University of Madrid, 28040 Madrid, Spain
2 School of Social Sciences, Yachay Tech, Urcuqui 100115, Ecuador; ernesto.pb@gmail.com
* Correspondence: sbelli@ucm.es

**Abstract: Purpose:** The purpose of this article is to explore how information and communication technologies have affected the way in which research groups are connected in their professional setting. It also analyses the techniques of interaction and cooperation of research groups, the exploration of information and the gateway to information, as well as the main tendencies and variations that researchers have observed during the last years. **Design/methodology/approach:** A survey was proposed to 305 scholars from different areas observing 159 variable responses. In order to complement and correctly interpret the information gathered within the survey, selected researchers were interviewed, and some of their opinions are reported in this work. **Findings:** Researchers see how their scientific networks expand thanks to digital tools, establishing professional relationships with other researchers. Moreover, researchers want to increase the amount of their scientific collaboration but they also need analogical meetings with others researchers, where face-to-face relations can accomplish the lack of this in digital communication. **Originality:** The dominance of tools used in scientific network activity is mixed, composed of analogical and digital ones. Researchers recognize the primary role assumed by digital practices in their work, increasing the quantity and the quality of it thanks to the efficiency and the access to information from different parts of the world. **Social implication:** Digital transformation has modified the research practices of researchers, from communication among them to the management of data. This new context of professional relations opens innovative implications and approaches for sharing knowledge between researchers and international collaborations.

**Keywords:** scientific networks; digital transformation; professional communication; collaboration; knowledge

## 1. Introduction

Nowadays our lives have introduced digital practices in all daily activities. This change is also reproduced in professional settings, such as scientific ones. In every scientific discipline, the so-called "digital turn" is observed. This term identifies a series of new professional practices to access and share knowledge among researchers. Digitalization has opened up new possibilities for researchers in their handling of information and knowledge. The suggestion that professional activities are being eroded by the proliferation of extensive 'networks' of professional possibility is implicit in Bauman's idea of 'liquid modernity'. Drawing a parallel with the concept of 'liquid love' (Bauman 2003; Hobbs et al. 2017), the concept of 'liquid science' can also be presented, arguing that it has transformed scientific activities into a type of practice where researchers can do research with colleagues from different parts of the world.

The production of knowledge and the development of technologies are fundamental to respond to social demands since they favor the availability of scientific resources to understand reality and act upon it. Thus, access to academic production becomes an essential element for social development, as it allows knowledge of research and technologies,

the development of new studies based on existing knowledge and to act on social reality consistently.

Unlike previous generations, current scholars have a range of possible academic collaborations and networks available through digital instruments. The Internet has become a dominant social and professional intermediary (Hobbs et al. 2017). Less time and effort are necessary to start and achieve academic cooperation. The position of physical universities, such as laboratories, offices, or classrooms, has been partially displaced, increasing scientific productivity through the internet-mediated collaboration (Duque et al. 2005, 2009; Hine 2006; Olson et al. 2008). Thanks to this study, it is possible to understand how networks in other contexts, such as professional, are influenced by digital technologies that modify the relations among participants.

The objective of the article is to explore how information and communication technologies in this specific environment have affected how academic groups are connected. In particular, this study explores how scientific networks develop and how they obtain and manage information through digital transformation. It also observes the techniques of communication and cooperation among research groups, the access to information and the main tendencies perceived in the last decade.

For this purpose, a mixed-methods study involving a survey and interviews was proposed to researchers. The scholars who were consulted embrace different areas and are situated in different countries. What follows is a review of the literature, the methodology and then an analysis of emerging patterns of academic practices.

## 2. Literature Review

Numerous bodies of literature inform this study. The first is the research on digital transformation of intimacy by Hobbs et al. (2017), which has inspired this paper. The authors explored the experience of users of digital dating apps to assess the extent to which a digital transformation of intimacy might be under way. They studied how apps are mediators through which persons engage in tactical acts in search of love, sex and intimacy in the so-called "networked intimacy". As noted earlier, drawing a parallel with the concept of "liquid love", Bauman (2003) says that virtual relationships are progressively replacing more static 'real' relationships. Moreover, the author states that the extensive practice of mediated communication is leading people to think more of passing relations than life-long partnership. In previous works, emotions related to information and communication technologies (ICT) in these transient connections have been studied (Belli et al. 2014), but it is not the purpose of this paper to analyze the emotions generated in scientific networks.

The second point is the research on digital transformation conducted in the project EULAC Focus[1], where it was observed that scientific networks are increased through the usage of digital technology. This increased the number of scientific collaborations and scientific production (Santoro and Belli 2018; Minniti et al. 2018). Santoro and Belli (2018) analyzed the use of digital technologies, its functions, limitations and potentialities of the academic community in Latin America, with the purpose of building a common framework that allows the mediation of technologies in the production of information and the development of equitable collaborative networks. For example, to incentivize a common open access policy for Latin America.

The third point of this review section is the variation of space where study is conducted. Scientists work in online and offline spaces at the same time. This organization is also called "e-Infrastructure" and it denotes the research ecosystem in which all scholars have shared access to unique or distributed scientific amenities regardless of their position in the world (Candela et al. 2011; Corti and Fielding 2016). Renaud (2000) defines infrastructure as the cooperative structures that allow, improve, express and structure research, incorporating data, facilities, tools and practices that enable communication within the community. Wellman (2001) says that with modern digital technology, the 'place' where we are may be constructed more from our ability to communicate with others than from our physical location. Digital data, information and networks allow individuals to decrease the

obstacles of location, time and discipline (Atkins et al. 2003). Such settings allow scientists to distribute and cooperate over time and geographic, organizational and disciplinary distance.

Studies that reflect emerging researchers' pre-Internet network shapes that they have been structurally isolated from the global scientific production (Gaillard 1991; Shrum 2005). Collaboration in science has converted progressively significantly due to scientific networks and a common vision that important challenges can only be focused on through research collaboration. In the same way, Alcaide and Ferri (2014) discuss that the upcoming challenges of scientific investigation are in forming fruitful alliances that can benefit research growth in less industrialized countries. Regional collaboration is recognized to be particularly significant for countries whose research organization can be helped by structural alliances with academics from universities abroad. Studies of cooperation in developing countries are of particular importance because initiatives are often the consequence of "research-for-aid" arrangements, based on North–South asymmetries (Chinchilla-Rodríguez et al. 2018). Issues conditioning the exercise of research in "peripheral contexts" can contain the choice of scientific disciplines and standards behind cooperation (Vessuri 1984; Kreimer 2000, 2006).

The Internet is firmly entrenched in the daily lives of millions of people, faithfully reproducing existing patterns of power inequalities across the globe, including critical differences in access to the Internet itself (Castells 2006; Haythornthwaite and Wellman 2002). ICT and technology transfer can benefit and move a country forward in economic and social development.

Numprasertchai and Igel (2005) observe that research units in developing countries have many disadvantages compared to newly industrialized countries and developed countries regarding based knowledge, experts, researchers and infrastructure. Inequality of access to information and technological advantages among researchers became a crucial factor in science. Many researchers, for one reason or another, cannot participate in the digital-mediated scientific networks. The implications of digital transformation need to be understood more deeply because of the way in which innovations in digital technologies are contributing to the exercise and distribution of power in society (Silverstone 1999). ICT is a suitable and effective way to manage knowledge, and has become an essential element to improve scientific performance (Chataway and Wield 2000; Garavelli et al. 2002; Prez-Bustamante 1999).

The knowledge deficiencies of less developed countries have been framed by documenting the geographical network configurations and geo-social inequalities found in the scientific communities located in resource-poor regions (Duque et al. 2009). There is rarely a network configuration that is balanced between both local (more likely to be a strong tie) and global (more likely to be a weak tie) contacts (Duque et al. 2009). This balance is considered the optimum social network configuration, but in the less developed world, the result is often a disjointed knowledge community, part of which is isolated from global science and part of which is highly dependent upon institutions located abroad (Duque et al. 2009). The suggestion is that these scientists' network profiles tend to favor either local or global ties.

Digital transformation has changed the way in which the academic community communicates. Scholars nowadays need to be experts in communicating their research, and increasing visibility on the Internet is a key success factor. Visibility is defined as the extent to which a user is likely to come across a reference to a Web site. Liquid science has increased collaboration between institutions and disciplines creating a digital infrastructure. Our concept of liquid science, composed of these contribution blocks, covers new forms of sharing knowledge among researchers (and society) as a relevant indicator of the emerging patterns of cooperation and negotiation, composed by the actors involved, their relations and knowledge, based on the model of Heimeriks et al. (2003).

## 3. Methodology

### 3.1. Design

This is a mixed methods' research involving a survey and interviews. The survey was initially shared to a multi-stage sample of American and European scientists in all areas from the accounts of the members of the EULAC Focus project to their research colleagues.

The invitation was then distributed via email and social networks (Twitter, Facebook, LinkedIn, ResearchGate, Academia.edu) in a 'snowballing' fashion. Although the 'snowball method' can have limitations with respect to producing statistically significant representative samples, this exploration method is capable of gathering data indicative of broader social patterns and trends, particularly when the survey spreads a broad cohort of contestants (Atkinson and Flint 2003; Denscombe 2010; Hobbs et al. 2017; Neuman 2011). They are "high performance" groups, in the sense that they are prestigious in their field, have financial resources and institutional support, and this translates into positively recognized results in scientific terms.

The population study covers scientists. The questionnaire included objects involving digital transformation, detailed queries about digital communication, e-science and open access, in addition to questions about their scientific practices. This methodology contains a mixture of descriptive and inferential analyses founded on the data gathered. For the inferences, the Poisson regression model was used, in its variant that modifies the over dispersion existing in the data, quasi-Poisson (Agresti 2015). This model has a clear application when the variable response is the result of situation including and pursues the determination of rates related with a period of time. All information processing was made in R Statistical Software (R Core Team 2018).

The objective was to analyze in what way the information and communication technologies have changed the manner in which scientists are related through 159 variables. For this analysis, the variables under consideration are in Figure 1. The survey contained of a mixture of open-ended, multiple-choice and Likert-scale questions and took approximately 15–20 min to complete.

| Code | Question | Var |
|------|----------|-----|
| P004 | Country: | Country |
| P006 | Gender: | Gender |
| P007 | Age: | Age |
| P012 | Do you primarily work alone or as part of a research/project team? | AloneTeam |
| P013 | How many people are there in these teams (typically)? | TeamSize |
| P014 | Where are team members located? [Same institution] | LocSameIns |
| P015 | Where are team members located? [Different institution] | LocDiffIns |
| P016 | Where are team members located? [In other Countries] | LocDiffCou |
| P017 | How often do team members interact (typically)? | FreqInt |
| P018 | How do team members keep in touch? Could you indicate whether the following are 'Essential', 'Used' but not essential, or 'Not Used': [Letters] | ContactLetter |
| P019 | How do team members keep in touch? Could you indicate whether the following are 'Essential', 'Used' but not essential, or 'Not Used': [Phone calls] | ContactPhone |
| P020 | How do team members keep in touch? Could you indicate whether the following are 'Essential', 'Used' but not essential, or 'Not Used': [E-mails] | ContactEmail |
| P021 | How do team members keep in touch? Could you indicate whether the following are 'Essential', 'Used' but not essential, or 'Not Used': [Discussion mailing lists] | ContactList |
| P022 | How do team members keep in touch? Could you indicate whether the following are 'Essential', 'Used' but not essential, or 'Not Used': [Instant messenger (WhatsApp, Line, etc.)] | ContactInsMes |
| P023 | How do team members keep in touch? Could you indicate whether the following are 'Essential', 'Used' but not essential, or 'Not Used': [Videoconferencing (Skype, GTalk, etc.)] | ContactVideoC |
| P093 | How often do you publish? Could you indicate the approximate average number of publications per year of the following types? [Journal Papers (not-Open Access)] | pubTotal, pubScience |
| P094 | How often do you publish? Could you indicate the approximate average number of publications per year of the following types? [Journal Papers (Open Access)] | pubTotal, pubScience |
| P095 | How often do you publish? Could you indicate the approximate average number of publications per year of the following types? [Books (Written or edited)] | pubTotal, pubScience |
| P096 | How often do you publish? Could you indicate the approximate average number of publications per year of the following types? [Book Chapters] | pubTotal, pubScience |
| P097 | How often do you publish? Could you indicate the approximate average number of publications per year of the following types? [Reports] | pubTotal, pubScience |
| P098 | How often do you publish? Could you indicate the approximate average number of publications per year of the following types? [Conference papers] | pubTotal, pubScience |
| P099 | How often do you publish? Could you indicate the approximate average number of publications per year of the following types? [Non-refereed articles] | pubTotal |
| P100 | How often do you publish? Could you indicate the approximate average number of publications per year of the following types? [Blog posts] | pubTotal |

**Figure 1.** The variables under consideration for the analysis.

We have observed the size of the researchers' networks (codes P012-13), its structure (local or global) (codes P014-016), the frequency of interactions (code P017), the modality of interaction (analogical or digital tool) (codes P018-023) and scientific productivity (codes P093-100). We controlled that the information that the survey respondents entered was reliable by matching their responses with their CVs that appear on institutional web pages and their indexed publications in Scopus and Web of Science. In this way, we evaluated if they participated in international projects and published articles with other authors. Although this system does not fully guarantee work as a network researcher, it allows us to understand if these scientific collaborations are successful or not and if they are translated into observable products such as scientific papers.

Different profiles were defined to be identified through the specific responses given by the interviewed sample:

- Analogic-Mixed–Digital researcher type: Dominance of the tool used in the scientific network activity;
- Passive–Active researcher type: Type of involvement in digital practices (good mood and bad mood);
- Low Scientific Activity–Medium Scientific Activity–High Scientific Activity: Level of scientific activity (local or global), observing frequencies of interactions and number of active scientific projects and publication involvement.

The approval for the research was given in September 2016.

### 3.2. Human Ethics

It is the responsibility of the EULAC Focus consortium that no harm whatsoever occurs to individuals by virtue of their participation in project activities and data collection processes. In overall terms, the research complies with the following ethical standards for research in social sciences and not involving greater than minimal risks for the subjects:

- The research is designed and applied ensuring transparency and integrity;
- The participants in surveys and interviews will be informed about the project objectives, methods and intended use of the collected data;
- The information gathered with the participation of human subjects will be treated confidentially and the anonymity of respondents will be respected;
- The participants will take part in the research voluntarily.

Regular project meetings and other checks by project leaders and the coordinator shall guarantee the compliance with ethical standards before publicizing any research findings. Finally, the EULAC Focus project will make an effort to pursue the integrative concept of Responsible Research and Innovation (RRI) and of Protection of Human Subjects (PHS).

### 3.3. Data Gathering

The first answer to the survey was recorded on 16 December 2016 and the last on 13 February 2018. Scientists belonged to 18 European and American countries. The survey had a total of 305 respondents, most of all (264) replied all of the questions under consideration. Only the replies of those interviewed who were working alone, and in a team or only in a team were considered. In terms of gender, 4 persons preferred not to inform (1%), whereas 101 were female (34%), and 200 were male (66%). Figure 2 displays the demographic pyramid of the survey.

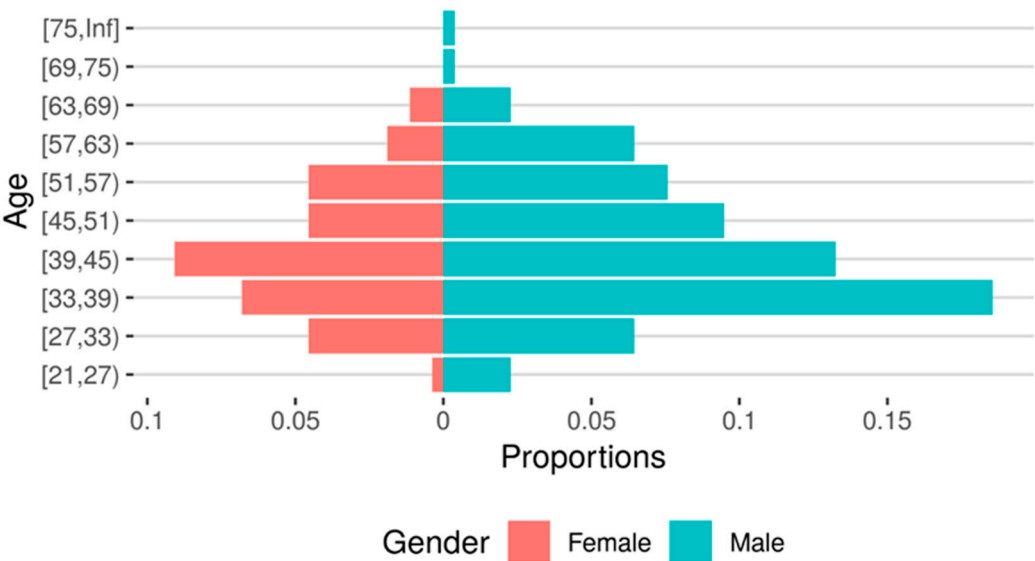

**Figure 2.** Demographic pyramid of the sample.

Figure 2 displays that those replying to the survey are mostly male, with ages between 33 and 45 years. Female researchers face 'patrifocal' constraints in some emerging world cultures that limit their entrée to new technologies and the aptitude to network outside the local arena (Palackal et al. 2006). Age is a social aspect that differentiates entrée to social networks that generally favor youth. In combination, unequal access to network and technology resources along with regional, gender and age dimensions can be gathered under the phrase 'geo-social asymmetries' (Duque et al. 2009).

Detailed demographic data was gathered from scientists as it is one of the main objectives of the EULAC Focus project. Table 1 and show that scientists from 18 different countries were surveyed and their areas of knowledge. In all, 67% of the respondents work in an Ecuadorian university, whereas 14% work for Mexican universities, and 5% work for Spanish universities. There is a prominence of Ecuador, because one of the researchers and author of this research is based in this country. The rest of the countries represent 13% of the sample. For regions, we have 1% from the USA, 9% from the European Union and 90% from Latin American Countries.

**Table 1.** Distribution of the sample for countries. 'f' refers to the absolute frequency, 'F' to the cumulative absolute frequency, 'rf' to the relative frequency (or proportion) and 'rF' to the relative accumulated frequency.

| Country | f | F | rf | rF |
|---|---|---|---|---|
| 01. Ecuador | 178 | 178 | 0.670 | 0.670 |
| 02. Mexico | 38 | 216 | 0.140 | 0.820 |
| 03. Spain | 13 | 229 | 0.050 | 0.870 |
| 04. Italy | 6 | 235 | 0.020 | 0.890 |
| 05. Argentina | 5 | 240 | 0.020 | 0.910 |
| 06. Brazil | 4 | 244 | 0.020 | 0.920 |
| 07. Cuba | 3 | 247 | 0.010 | 0.940 |
| 08. USA | 3 | 250 | 0.010 | 0.950 |
| 09. Chile | 2 | 252 | 0.010 | 0.950 |
| 10. Colombia | 2 | 254 | 0.010 | 0.960 |
| 11. Peru | 2 | 256 | 0.010 | 0.970 |
| 12. Venezuela | 2 | 258 | 0.010 | 0.980 |

**Table 1.** *Cont.*

| Country | f | F | rf | rF |
|---|---|---|---|---|
| 13. Czech Republic | 1 | 259 | 0.000 | 0.980 |
| 14. France | 1 | 260 | 0.000 | 0.980 |
| 15. Germany | 1 | 261 | 0.000 | 0.990 |
| 16. Netherlands | 1 | 262 | 0.000 | 0.990 |
| 17. Portugal | 1 | 263 | 0.000 | 1.000 |
| 18. UK | 1 | 264 | 0.000 | 1.000 |

The majority of researchers work in universities (92%), where they are professors (68%), graduate students (13%), and post docs (5%). Their specialties are shown in Table 2.

**Table 2.** Specialties of the interviewees.

| Specialty | f | F | rf | rF |
|---|---|---|---|---|
| 01. Technological Science | 43 | 43 | 0.160 | 0.160 |
| 02. Life Sciences | 35 | 78 | 0.130 | 0.300 |
| 03. Mathematics | 20 | 98 | 0.080 | 0.370 |
| 04. Medical Sciences | 16 | 114 | 0.060 | 0.430 |
| 05. Pedagogy | 15 | 129 | 0.060 | 0.490 |
| 06. Sociology | 15 | 144 | 0.060 | 0.550 |
| 07. Agricultural Sciences | 14 | 158 | 0.050 | 0.600 |
| 08. Economic Sciences | 14 | 172 | 0.050 | 0.650 |
| 09. Physics | 14 | 186 | 0.050 | 0.700 |
| 10. Science Of Arts And Letters | 13 | 199 | 0.050 | 0.750 |
| 11. Science Of Earth And Space | 13 | 212 | 0.050 | 0.800 |
| 12. Geography | 11 | 223 | 0.040 | 0.840 |
| 13. Astronomy And Astrophysics | 10 | 233 | 0.040 | 0.880 |
| 14. Chemistry | 9 | 242 | 0.030 | 0.920 |
| 15. Politic Science | 7 | 249 | 0.030 | 0.940 |
| 16. Psychology | 7 | 256 | 0.030 | 0.970 |
| 17. Other | 3 | 259 | 0.010 | 0.980 |
| 18. History | 2 | 261 | 0.010 | 0.990 |
| 19. Science (Field Not Defined) | 2 | 263 | 0.010 | 1.000 |
| 20. Earth Sciences | 1 | 264 | 0.000 | 1.000 |

Survey participants were self-selected to contribute in a follow-up in-depth, semi-structured interview, and recording the interview in person. The interviewees included 39 researchers aged between 32 and 62 years, residents in Ecuador, Mexico, Uruguay, Spain and Italy.

The preponderance of the in-depth interviews was conducted in researchers' offices and laboratories in 2017 and 2018. To preserve researchers' anonymity, they were given pseudonyms in all transcripts. The interviews sought to cover more study topics than appeared from the survey, including different ways to connect with their partners, search information, publishing papers and their everyday activities to do research in their environment.

We used a Content Analysis as it is considered a good instrument to analyze qualitative data (Dikmenli 2009; Kaleli-Yilmaz 2015). This method allows the separation of the information into its components and it studies them for comparisons or shared factors. Dikmenli (2009) showed that this method offers reliable results in many studies (Hirvonen and Viiri 2002). Once all the interviews were transcribed, potentially relevant segments for the research objectives were identified. Thus, in the first phase of analysis, an open code identifying thematic blocks on production practices and scientific communication was established. Subsequently, different subtopics assigned to each thematic area were identified.

Several matrices were elaborated to include the different types of data collected. Matrices related to the relations among items with participants and shaped for each wave individually using the analogous ethnographic information. When an item is used by a research member or when an informant mentioned that he or she used an item, a link between the participants and the item was listed. These results were triangulated with the results of the surveys, based on video interactions and interview answers about links with other researchers.

Then, a conceptual diagram was elaborated (with the help of Atlas.ti software, version 8) and condensed the information of each topic and subtopic from the compilation of the statements made explicitly by the people interviewed, limiting the interpretation of implicit connotations to comments added in the form of external notes and thematic groupings. The findings are presented in the next part through their qualitative and quantitative dimensions.

## 4. Analysis

### 4.1. Networks or Not Networks

In our analysis, we focus the attention on the research networks and in what way these networks work, examining the structure of the relations between the scientists and the usage of different technologies. We agree with Bruno Latour (1988) that the construction of these links should not be understood as purely symbolic or abstract, but rather as something material (such as physical infrastructure, laboratories, offices, tools and computer networks). Scientists and analogical/digital items establish this networked academic community where everything is continuously in mediation, cooperation and collaboration.

The first question of the analysis is how researchers work: individually or collectively, alone or in a research team. In total 49% of them ($n = 150$) prefer to work as part of a scientific group, and just a minor number of academics, 6% ($n = 19$), work totally alone. In total 45% ($n = 136$) of the participants prefer to work in both ways (collectively and alone). The dimension of their academic team is between three and six researchers (63%, $n = 191$). Only 34% ($n = 80$) of the scientists work in teams that have more than six members (see Figure 3).

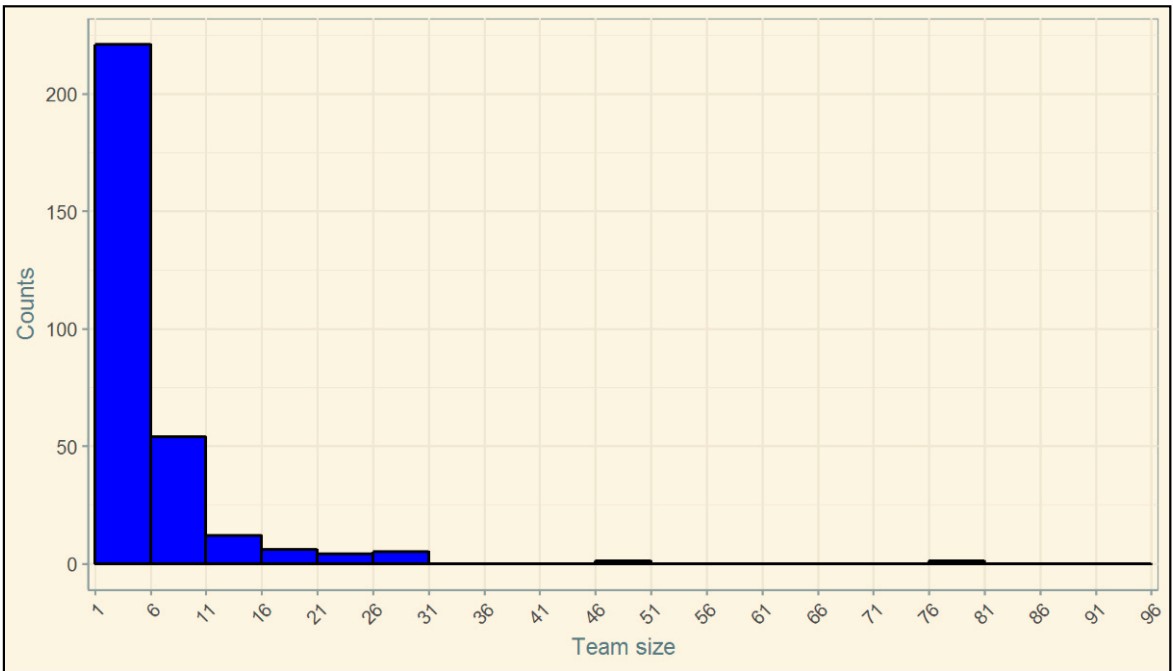

**Figure 3.** Size of the research groups.

Once the size of these teams is defined, it is important to understand where the researchers are located and in which institution or country they work. The majority of their networks are situated in the same organization (36%), 24% of the researchers have partners in the same country and 20% of the sample report colleagues from different countries. These percentages are helpful to understand the allocation of scientific groups, and to distinguish if these researchers are part of international or national research groups.

Secondly, it is observed how these scientific networks work and how researchers create and manage links among them. Researchers continuously share information and data. This flowing knowledge is the basis of scientific research, as observed by Latour (1988). In Table 3, the interaction incidence of team members can be detected.

**Table 3.** Interaction frequency of the team members.

| FreqMeet | f | F | rf | rF |
|---|---|---|---|---|
| 1. Weekly | 111 | 111 | 0.37 | 0.37 |
| 2. Twice a week | 69 | 180 | 0.23 | 0.59 |
| 3. Monthly | 56 | 236 | 0.18 | 0.78 |
| 4. Daily | 39 | 275 | 0.13 | 0.90 |
| 5. Bi-weekly | 17 | 292 | 0.06 | 0.96 |
| 6. Less often | 12 | 304 | 0.04 | 1.00 |

The majority of researchers communicate between themselves either one (37%, n = 111) or two times (23%, n = 69) per week. Only 18% (n = 56) of the surveyed interact one time per month, and 13% once per day.

Concentrating on the type of instruments to interconnect with partners, we can perceive in what way these tools have transformed over the last few years, and if they have passed from analogical to digital instruments to converse with the rest of their partners or not. Scientists connect with the rest of the nodes of their networks by using analogical (Letters, Phone Calls) and digital items (Emails, Mailing Lists, Instant Messaging, Videoconference). It can be perceived that letters are not used by 84% (n = 256) of the scholars, transforming it into an obsolescent device. The majority of scientists (80%, n = 244) agree that emails are crucial in their research activities.

Over the last few years, messaging or instant messaging services have amplified their influence on communication among partners. For scientists, this facility is considered indispensable and helpful (89%, n = 269). This informal tool is important for daily conversations and helps to have a response very quickly. Many of these messaging services are available on smartphones and can be used while the researcher is not in the laboratory or office. Many times, these tools are used in informal settings, while the researcher is drinking a coffee in a public place or traveling on the train, such as J. explained, "While I take a coffee in the university I used to converse with my research partners in different parts of the planet via WhatsApp or Telegram. If they are in the same country I call them directly with the phone."

Even if the interviewees mentioned different contexts to communicate with their partners, email is still one of the best popular means on the Internet in science. In scientific collaboration, discussion groups and mailing lists transfer knowledge daily between researchers and are a valuable item to obtain access to knowledge and information.

Moreover, social networks remain an important, and informal, channel of communication between researchers. As G. points out, "Usually, Facebook or Twitter are for private communication, Academia and ResearchGate are more specific for my work." These platforms allow researchers to create their professional identity following a dynamic that encourages other researchers to read their work or collaborate with them.

Communication technologies widely used for informal communication, such as Skype, WhatsApp, and email, are taken as necessary and indispensable tools for scientific practice since they allow the establishment of new contacts and the maintenance of links among researchers, increasing the number of scientific collaborations and production (Santoro and Belli 2018). Many of these tools, although they were created for personal conversations, have found a new scope in professional settings. Thanks to their informal dynamics and massive presence in society, they facilitate a faster and more flexible communication by building the new context for Liquid Science. As J. said, "I think email is a hundred percent necessary ( . . . ) Social networks play a fairly important role for my scientific relationships. But through Skype, WhatsApp, among others, we can send large amounts of project data and coordinate activities, meetings, etc. efficiently and, most importantly, instantly, since time is very valuable in science. There are other social networks such as LinkedIn or ResearchGate that are very useful when trying to find colleagues to work or to find new projects or even to download documents from people related to your field." In this extract, J. shows how in the last decade social networks have taken the spotlight in interpersonal communications. Therefore, massive social networks are very appealing due to their facility of usage and appropriateness for contemporary scientific practices.

One of the tools that we have not considered in the survey, but that represents an important tool to communicate with partners in the e-infrastructure (Candela et al. 2011; Corti and Fielding 2016) is the Cloud. This tool appeared in different semi-structured interviews. As S. points out, "These are spaces where basically share documents for working together, where often a dialogue is generated with the rest of the colleagues through comments and the chat window." The Cloud is the most surprising revolution in the last decade, because it is a manner of organizing collaboration (Brunswicker et al. 2017) as L. puts it, "Before I used emails with text and the track changes activated. Now, I use the cloud to sync data within the team and create a backup of old files, using a cooperative tool to work instantaneously on documents."

It is very interesting for the purposes of this research, how an "old" and analogical tool, such as the phone, is considered essential for many researchers. Personal telephone conversations between researchers remains one of the most important tools to communicate. As Goffman (1983, p. 3) states, "A phone call affords a degree of mutual monitoring, warrants focused attention and facilitates the 'sustained, intimate, coordination of action' typical of verbal conversation." As the researcher G. from the Geology area points out, "Yes, it is very important how we communicate with the colleagues, but the most important thing is that communication can funnel in the best fitted tool to calibrate it. I told you why, the solution that we found to solve intense storms problem was to use a technology that everybody has: the phone." We have observed that a researcher who does not use the phone to contact colleagues (versus the one who considers it essential), decreases the number of scientific publications by 32.83%. The same situation occurs for a researcher who uses the phone to contact colleagues but does not consider it essential (versus the one who considers it essential), which decreases the number of scientific publications by 27.83%.

Videoconferencing can be considered the digital transformation of the phone. Many researchers miss face-to-face conversations in their networks. For this reason, videoconferencing represents a good "surrogate" to see the faces of their international colleagues, and to maintain long online meetings. Videoconferencing has changed radically the way to conduct research, and in many cases is the most complete tool to work in groups, as the researcher C. said, "Skype. I speak with them once a week or once every two weeks, and by email daily." Likewise, the potential of videoconferencing to connect people in different spaces and times facilitates coordination in the development of collaborative works. This tool increases efficiency between partners. However, many times it also generates negative emotions caused by cultural differences and technical problems (Belli 2018).

A researcher who do not use videoconferencing to contact colleagues (versus the one who considers it essential), decreases the number of scientific publications by 49.97%. Instead, a researcher who uses videoconferences to contact colleagues, but does not consider it essential (versus the one who considers it essential), decreases the number of scientific publications by 23.29%. Consequently, videoconferences are considered an essential tool to communicate between researchers.

Moreover, videoconferences offer a plus to researchers, as H. said, "We interact by Skype because we can share data and ideas through the screen." Videoconferences have changed radically how researchers work, as many of them claim. Different platforms to have online meeting exist, but many times researchers prefer Skype because it is the most common tool. As P. points it, "For instance, I use Skype for conferences, like I contact with normal phone calls different people. Also, when I have to give a talk, and my colleague insists that I must use another software, I can't remember the name, I continue to prefer Skype, because it is easier." V. points out as well, "Of course, for example we use for our meetings tools like Skype, Zoom and Google Hangouts, but mostly is Skype."

We have observed in this section how phone and videoconference calls are a vehicle to communicate in a semi-direct way among researchers to achieve specific scientific aims. Researchers share the same time-frame, and the interaction is closer to a real face-to-face conversation than email exchanges, where there is no mediated co-presence and dialogical negotiation (Muntanyola and Romero 2014).

### 4.2. Positive and Negative Aspects of the Digital Transformation of Science

In this part it will be analyzed how collaboration in science has changed thanks to the e-infrastructure, and the effects of this digital transition. For 59% (n = 179) of those surveyed, digital science has changed their ways of collaboration, while 17% (n = 53) assume that it has changed, but not much. A total of 24% (n = 72) of the researchers surveyed agree that the apparition of the online information environment has not changed the way to cooperate with partners in research.

The survey also considered the problems related to new forms of communication using digital technologies. For 54% (n = 117) of the researchers no problems exist related to this; 18% (n = 39) consider that problems may exist, and for the rest of the respondents (27%, n = 59) there are real problems with these new forms of communication. For the respondents, these problems are not related to the tool, but how that tool is used.

In Table 4, we have summarized the positive and negative aspects that respondents have considered about the digital transformation of science divided in three categories: Efficiency, Information Flow and Communication at Work.

**Table 4.** Positive and Negative Aspects of Digital Transformation.

| The 3 Main Changes of the Digital Transformation | Positive Aspects | Negative Aspects |
|---|---|---|
| **Efficiency** | - Easy and fast way to communicate<br>- Organize tasks in a quickly way<br>- Quick responses<br>- Scientific productivity increased | - Too much multitasking<br>- Research time is not compatible with "internet time"<br>- No interruption between private and professional life<br>- Dependency |
| **Information Flow** | - Access in every place and every moment<br>- Always updated | - Many times, it is of low quality and superficial<br>- Too much information and not too much time to revise |

**Table 4.** *Cont.*

| The 3 Main Changes of the Digital Transformation | Positive Aspects | Negative Aspects |
|---|---|---|
| **Communication at Work** | - It is direct and without barriers between North–South collaborations<br>- Based on a horizontal architecture<br>- No hierarchies | - Impersonal way<br>- Lack of face to face interactions<br>- Difficult to share emotions<br>- Causing misunderstandings<br>- No real interaction<br>- Need an extra-time to arrive to a consensus<br>- Lack of formality<br>- Negative emotions caused if there is a delay in answering |

The more enthusiastic affirm that digital transformation has changed the way of collaboration, increasing efficiency because, although the colleagues are not physically close, they have access to the same information. Half of the answers collected are about how efficiency has changed in their work with colleagues, as F. replies, "It is possible to organize and manage tasks between researchers in a quickly way", or J. claims, "Scientific collaboration becomes faster, and thanks to cloud computing it is possible to work together in the same document at the same time." As these two researchers point out, this type of remote collaboration offered by the digital transformation optimizes the resources as geographical distance among researchers is not an issue anymore. Researchers can also obtain quicker responses from the scientific community in cases of emergency, and it allows researchers to be more specific with data, increasing productivity. Digital infrastructure can support a better exchange of ideas among researchers and, consequently, scientific productivity increases.

Researchers have experienced a different temporality in their work hours and in their colleagues' work hours, with the result that priorities of each are not respected within scientific networks, generating negative emotions. Some researchers feel the impossibility of disconnecting from their networks, which is a sort of dependency of their work, and the use of these digital tools in their lives appears to be too excessive. In many cases, researchers do not approve the increase in work alcoholism in many research groups. Researchers have also experienced a difficulty in managing their time to do research, the ambiguity between urgent tasks and non-urgent, and how to manage the quantity of mails, messages and documents that they receive every day.

The second category is the information flow between researchers. As D. points out, "Now, everyone can access this information immediately, shared through the cloud and always 'fresh'." In huge and international research projects, it is important to use these digital tools to share information to be competitive, pursuing multiple scientific international networks simultaneously. Science is not a physical issue that only 'exists' in books and in laboratories, but it is distributed everywhere, as this researcher J. explained, "This allows to work independently from different parts of the world, in many remote zones, with the condition to have Internet access." Researchers have access to the information that they need immediately, changing the form with which they manage and shape knowledge.

For some researchers, in these digital scientific interactions, there is not a high-quality production of knowledge, because everything is fast and superficial. There is often not much time to focus on knowledge in these international scientific networks or to go deeper into research. The lack of time to be dedicated to the processes of researching, reading and writing, reveals how researchers receive and accumulate too much information and not having enough time to manage it. There is a difficulty to manage and elaborate many information and data in a limited time, and the researcher can collapse. Moreover, sometimes researchers are too much distracted while they use digital tools, doing multitasking activities (Belli et al. 2014), as this researcher S. explained, "The time to do research is

not compatible with the time of Internet, because to do research is necessary a complex thinking with no distractions, considering variables, and observing these, it is also design of the research, data collection and reflection about results, as like article redaction." What is perceived as information overload may be work overload, where too much information may reflect too much to do (Bawden and Robinson 2009).

Some researchers (Kennedy 2001) have coined the term "Information anxiety" to describe a situation of stress caused by the inability to access, understand or make use of necessary information. The reason for this anxiety is the information excess or insufficient information (Bawden and Robinson 2009).

The third aspect of this digital transformation in science is communication. International collaboration based on communication between researchers is recognized to be particularly significant for countries whose research organization can be helped by structural alliances with academics from universities abroad. Studies of cooperation in developing countries are of particular implication based on North–South collaborations and conditioning the exercise of research in less developed countries. Digital communication has distorted geography by shrinking distances and removing access barriers.

Although this aspect has taken enormous advantages from this transformation, one of the most important issues is the lack of face-to-face interactions. Many times, for the respondents, it is not easy to make decisions and help to flow the research process. Digital communication produces an impersonal way of communicating, without emotions, or on the contrary, it is too subjective and causes misunderstandings (Sabando and Belli 2017). It is the same issue that happens in romantic relationships using digital communications when two persons try to maintain a distant relationship. Emotional aspects are modified through the digital tool. As there is a lack of 'human touch', no real interaction between people happens, as R. said, "Many times it is needed to have more time to arrive to a consensus, but we are living in fast times and we do not have this extra time." This lack of emotions does not allow negotiation or decision making a decision such as in face-to-face interaction. Some researchers experience a lack of formality in this type of communication, because digital tools are constructed by an "horizontal architecture" (Belli and Aceros 2016) where every participant has the same role and weight, forgetting in many cases the hierarchies and positions. It is not the same as in physical meetings, where hierarchies work and consensus is given by the authority in most of the cases due to a pyramidal structure. For these reasons, many researchers affirm that their participation in conferences, workshops, and scientific meetings is essential to maintain their networks and recognize their colleagues in a physical world.

One of the main problems of digital communication is the lack of interruption between professional and private life, as the possibility of being contacted at any time, and on weekends and holidays too always exists. It is hard to distinguish work hours and the rest of the day. This problem affects not only the recipient of the message, but also the sender. When the response is not immediate, negative emotions are generated in the sender, the researcher A. commented, "Now we are less patient and we need a quick response from our colleagues." The time limit to answering an email has been reduced in the last ten years (Belli et al. 2014), as D. said, "The email is a formal tool and it is normal to answer it in maximum two days. On the other hand, a WhatsApp message is instant but is less formal and the time of response decrease." Kraut et al. (1988) differentiated two types of communication. The first one is interactive communication, which involves the ability to exchange information rapidly and adjust message responses to communication partners. The second one is expressive communication, which requires the capacity to convey ideas considering not only semantic but also contextual meaning (body language, intonation, background, etc.). We have observed how many researchers mixed both types of communication, causing problems in research activity and in their scientific networks. In many informal platforms, messages are not clear, and they lack information and have errors due to the low cost to write and send them. We have observed that if the tool is more

formal, more time is needed to respond. On the other hand, if the tool is less formal, less time is needed.

## 5. Discussion

Thanks to the analysis, we have observed how the digital transformation has modified the professional practices of researchers to collaborate. The behaviors that mutual physical embeddedness of researchers have accomplished contrast with those of cooperative relations, which clearly embody themselves in the material world through sharing items in collaborative work depending on the task, the area of research and the position of the researcher. Digital technologies not only improve the effectiveness of scientific work but also allow the availability of digital scientific infrastructures. For example, we have access to the equipment or the performance of simulations in institutional contexts where there is a lack of technology, as in less developed countries where a large part of this research was conducted. In this sense, they express that those digital technologies serve as a patch for situations in which state-of-the-art technologies are not available (duly equipped laboratories, computer programs, etc.) in the institution or country in which the research is carried out.

The role of technology in researchers' workplaces, as we observed, is actively addressing against the centralization of the objects used, and it has a strong predisposition to closure in physical context networks. Screens, books, microscopes and so on are considered as everyday analog objects for researchers from different disciplines. It is the same for some digital objects such as word processors, spreadsheet programs, communication tools, internet browsers and email mailboxes. Shared objects filled by researchers incline to be shared in their networks, and there is no propensity for any items to be more central than the rest. For the networks of item usage, some effects are found as well. Researchers achieve the use of different objects and to combine the objects they use. While pairs of researchers do happen to share objects, they refrain from focusing on some particular objects rather than the rest. In other words, physical engagement tends to spread similarly between researchers.

Following recent micro-sociological and cognitive science studies (Muntanyola and Belli 2016), knowledge and information are claimed to be the product of interaction within a professional team. We have observed several features of how liquid science flows in scientific networks thanks to our research. The digital transformation in science has changed and modified practices between researchers, influencing directly their research.

It is extraordinary that in the digital era, where communication is immediate and flows through displays, phone conversations are still considered indispensable to have successful interactions with partners. Although emails and instant messages are considered important too, phone conversations help researchers to negotiate and cooperate among colleagues without causing misunderstanding such as happens with messaging services. We have observed that the usage of this mode of communication helps to be more productive and increases the number of scientific publications and scientific collaborations.

The development of phone conversations is the videoconference, where facial expressions and body language help to enrich conversations. It also allows the sharing of documents and data in real time. Digital infrastructure helps scientists to allocate research facilities regardless of their location. In this new era, researchers need to incorporate soft skills and techniques to carry out their work using these digital tools.

The problem caused by digital transformation of science is the cost to access the specific digital tools and devices. Many services for research are paid for, and require extensive payment from institutions and researchers. The networked environment provides a tantalizing amount of possible information, but it puts too many financial obstacles in the way. In many parts of the world, especially in some less developed countries, it is a problem to access the Internet. Many researchers do not have access to this technology or do not know how to use the technology. Internet or servers many times fail, and it is not possible to access information or to connect for a videoconference. When there are technical difficulties (Belli 2018), researchers are not able to maintain the meeting for different reasons. ICT

and technology transfer can benefit and move a country forward in economic and social development. However, ICT and access to the Internet have been inequitably distributed in most developing countries. Regarding science production and dissemination, this divide articulates itself also through the inequitable distribution of access to scientific knowledge and networks, which results in the exclusion of developing countries from the center of science. Developing countries are on the fringe of science, technology and innovation production due to not only low investment in research but also the difficulties in accessing the international scholarly infrastructure, such as scientific databases. Technical problems are always present during digital meetings, internet connectivity is often not fast enough and users end up experiencing critical technical difficulties.

This finding is consistent with the argument advanced by Huang (2014) about the importance of interpersonal collaborative networks within organizations for capacity building. Huang (2014) used social capital and social network theories to describe the mechanisms whereby interpersonal relations may enhance research productivity and the capacity of individuals and organizations. The positive and negative aspects of the digital transformation of science conveys to us that the connectivity of a scholar in a research network is more essential than the quality of research. For Callon and Law (2004), circulation has become more crucial than fixed positions. Liquid science is this conjunction of communication, knowledge and networks; increasing collaboration between institutions and researchers creating a digital infrastructure (Brunswicker et al. 2017). We have observed that the tendency is to have scientists working in medium-sized (from three to six members) international research groups, communicating between them one or two times per week. The dominance of tools used in scientific network activity is mixed, composed of analogical and digital ones, considering that phone conversations move in both of these two worlds thanks to smartphones and their different digital calling apps.

Researchers recognize the primary role assumed by digital practices in their work, increasing the quantity and the quality of it thanks to the efficiency and access to information from different parts of the world. We consider this as an active position in relation to the digital transformation. Although negative aspects of this digital transformation in science exist, the positive ones are the most significant.

The main representative changes that digital transformation in research networks has afforded to the research community are three: (1) Efficiency; (2) Information and knowledge flow; and (3) Communication.

The empirical results proposed by this research recommend that scientists observe the digital transformation of research as a welcome intermediary to produce and administer research networks. Unlike the argument advanced by Bauman, digital tools and digital communication are not 'liquefying' face-to-face communication, physical meeting and conferences. Indeed, data suggest that most of the researchers come to value and seek these social phenomena, and are merely using technology as a means to purse professional collaborations. Researchers feel that they have more professional possibilities than previous generations, and that technologies give them more agency to pursuing and meeting possible academic partners. The concept of 'networked individualism' (Rainie and Wellman 2012) is appropriate to studies of scientific networks of researchers, as individuals become more responsible for their professional practices within an extended social network ecosystem. These digital tools provide a scientific network that enhances the researcher's activity in what it is called 'liquid science'. 'Liquid science' is permitting them to approach a wide network of scientific opportunities. These scientific networks enhance a researcher's capacity to build new relationships with researchers from other parts of the world. Liquid science brings new opportunities in the researcher's life.

Rather artificial and not specific, conventional views on the distinction among analogic and digital tools are still mainly predominant. However, the conclusions which can be drawn from our study should be partial for three reasons. Firstly, the lack of representatives of different countries, as the study for the survey was carried out mostly in only three of them and in-depth interviews were completed in five countries. These countries were

chosen according to the availability of scientists that cooperate with researchers of the EULAC Focus project. Consequently, further investigation is required. Secondly, while we applied a mixed methodology to reduce the limitations caused by quantitative and qualitative methodology, we did not accommodate all potential options or measure results produced by the different interview techniques (Vannieuwenhuyze et al. 2010). Thirdly, the study linked to the usages made by academics of diverse discipline and in diverse stages of their careers. These scholars can assume and shape uncommon perceptions of the digital instruments that they use. Lastly, the information does not permit checking if these research networks produce some positive outcomes to science, or if they are not competitive such as other research networks. The study has bias that can influence a scientist to reply in an optimistic way about the introduction of digital technology in their research practices and networks. It is assumed that the survey may induce a bias to a digital profile due to the topic of the survey. The methodology based on surveys via the Internet can produce a bias in the results of the research, because the researcher who is not yet connected through ICT cannot or will not participate

Some conclusions can be drawn from these findings. As was discussed before, some researchers see how their scientific networks expand thanks to digital tools, establishing professional relationships with other countries. Moreover, researchers want to increase the number of their scientific collaboration but they also need analogical meetings with others researchers, where face-to-face relations can accomplish the lack of this in digital communication. Thus, empirical evidence suggests that researchers do not avoid analogical communication within their networks. They desire to continue to move forward in this mixed world where they can take advantage of both situations promoting international collaborations.

**Author Contributions:** Conceptualization, S.B. and E.P.; methodology, S.B. and E.P.; software, E.P.; validation, E.P.; formal analysis, E.P.; investigation, S.B.; resources, S.B.; data curation, E.P.; writing—original draft preparation, S.B.; writing—review and editing, S.B.; visualization, E.P.; supervision, S.B.; project administration, S.B.; funding acquisition, S.B. All authors have read and agreed to the published version of the manuscript.

**Funding:** This study was funded by Comunidad de Madrid, Atracción de Talento modalidad 1 (grant number 2018-T1/SOC-10409).

**Institutional Review Board Statement:** The study was conducted in accordance with the Declaration of Helsinki, and approved by the Institutional Review Board of Complutense University of Marid (date of approval 01 February 2019).

**Informed Consent Statement:** Informed consent was obtained from all subjects involved in the study.

**Data Availability Statement:** For data, please contact the corresponding author.

**Conflicts of Interest:** The authors declare no conflict of interest.

## Note

1   For more information about the project EULAC Focus visit http://eulac-focus.net/ (accessed on 1 January 2020).

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
