# Peer review of "Liquid Science and Digital Transformation: How Knowledge between Researchers Flows in Their Scientific Networks"

_socsci, doi:10.3390/socsci11040172_

Round 1

Reviewer 1 Report

I read this paper with interest. I do study "networks", but more from the graph theory perspective or applied far more mathematically than in this paper. So please forgive me for my lack of detailed comments.

The paper reads well, there is plenty of discussion and citation of related work. The data are presented clearly.

My only minor quibble--but I think it's an important one--is that on page 10, in Figure 3, the ordering of the 4 labels is illogical: Essential is obviously the most important, followed by "used", then "not used", and "no answer". That is the order you should put the labels in vertically in the figures. Whether you go top to bottom or bottom to top is not important, but "Essential" and "used" should be side-by-side, since they both qualify as "used". Currently they are at opposite ends of the bar graph, which makes it difficult to figure out what fraction of respondents actually *use* each tool--whether they label it as "essential" or not.

Author Response

Dear Reviewer 1,

Thank you for your suggestion. I agree with it. For this reason, we have eliminated figure 3 because it does not offer important information about the use of the tools that can be described without using the table. Thank you.

All the best,

Reviewer 2 Report

Dear authors: some issues to be improved:

About the theoretical framework
The theoretical framework is based on reports with more than 10 years, some of them are 21 years old. The reality of ICT and current researchers cannot be described based on these data.
For example, this quote (page 115-:   -"countries: “… poor countries (and within them poorer segments of the population) are being further marginalized, as their access to opportunities for wealth creation is being reduced; considerable development opportunities are being missed, as productivity and efficiency gains are not being transmitted from rich to poor countries.” (Digital Opportunity Task Force 2001: 3)So digital transformation produces a digital divide between developed countries and least developed and emergent 120 countries, where access to the Internet and ICT is not distributed in the same way. The consequence ..“ Is this happening now? China 21 years ago was not at the forefront, however today Chinese researchers have the best technological means in the world. All the references of the theoretical framework should be updated and provide data from the last five years  

About the methodology:
The methodology based on surveys via the Internet can produce a bias in the results of your research, because the researcher who is not yet so connected through ICT cannot or will not participate. This question should be discussed and appear in the limitations.  

Kind regards  

Author Response

Dear Reviewer 2,

Thank you for these important suggestions.

We have eliminated old references and introduced the limitation that you have identified.

Many thanks,

All the best,

Authors

Reviewer 3 Report

The overall manuscript is neat with relevant information for existing literature. One aspect that can be improved is the writing more to the point. Some descriptions are odd. I would suggest to examine existing literature to come up with more common descriptions of sections. This would also allow your work to extend existing literature.

Author Response

Dear Reviewer 3.

Thank you for these important suggestions.

We have introduced more updated literature references to the paper.

Thanks,

All the best,

Authors